# The Reversible Residual Network: Backpropagation Without Storing Activations

**Aidan N. Gomez**[*1], **Mengye Ren**[*1,2,3], **Raquel Urtasun**[1,2,3], **Roger B. Grosse**[1,2]
University of Toronto[1]
Vector Institute for Artificial Intelligence[2]
Uber Advanced Technologies Group[3]
{aidan, mren, urtasun, rgrosse}@cs.toronto.edu

## Abstract

Deep residual networks (ResNets) have significantly pushed forward the state-of-the-art on image classification, increasing in performance as networks grow both deeper and wider. However, memory consumption becomes a bottleneck, as one needs to store the activations in order to calculate gradients using backpropagation. We present the Reversible Residual Network (RevNet), a variant of ResNets where each layer's activations can be reconstructed exactly from the next layer's. Therefore, the activations for most layers need not be stored in memory during backpropagation. We demonstrate the effectiveness of RevNets on CIFAR-10, CIFAR-100, and ImageNet, establishing nearly identical classification accuracy to equally-sized ResNets, even though the activation storage requirements are independent of depth.

## 1 Introduction

Over the last five years, deep convolutional neural networks have enabled rapid performance improvements across a wide range of visual processing tasks [19, 26, 20]. For the most part, the state-of-the-art networks have been growing deeper. For instance, deep residual networks (ResNets) [13] are the state-of-the-art architecture across multiple computer vision tasks [19, 26, 20]. The key architectural innovation behind ResNets was the residual block, which allows information to be passed directly through, making the backpropagated error signals less prone to exploding or vanishing. This made it possible to train networks with hundreds of layers, and this vastly increased depth led to significant performance gains.

Nearly all modern neural networks are trained using backpropagation. Since backpropagation requires storing the network's activations in memory, the memory cost is proportional to the number of units in the network. Unfortunately, this means that as networks grow wider and deeper, storing the activations imposes an increasing memory burden, which has become a bottleneck for many applications [34, 37]. Graphics processing units (GPUs) have limited memory capacity, leading to constraints often exceeded by state-of-the-art architectures, some of which reach over one thousand layers [13]. Training large networks may require parallelization across multiple GPUs [7, 28], which is both expensive and complicated to implement. Due to memory constraints, modern architectures are often trained with a mini-batch size of 1 (e.g. [34, 37]), which is inefficient for stochastic gradient methods [11]. Reducing the memory cost of storing activations would significantly improve our ability to efficiently train wider and deeper networks.

---

[*]These authors contributed equally.
Code available at https://github.com/renmengye/revnet-public

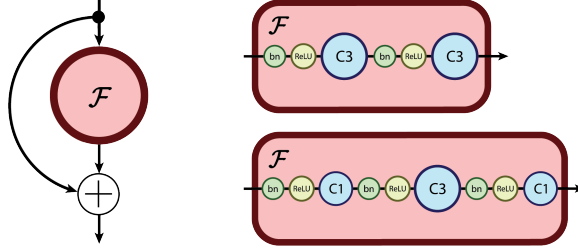

Figure 1: (left) A traditional residual block as in Equation 2. (right-top) A basic residual function. (right-bottom) A bottleneck residual function.

We present Reversible Residual Networks (RevNets), a variant of ResNets which is reversible in the sense that each layer's activations can be computed from the subsequent reversible layer's activations. This enables us to perform backpropagation without storing the activations in memory, with the exception of a handful of non-reversible layers. The result is a network architecture whose activation storage requirements are independent of depth, and typically at least an order of magnitude smaller compared with equally sized ResNets. Surprisingly, constraining the architecture to be reversible incurs no noticeable loss in performance: in our experiments, RevNets achieved nearly identical classification accuracy to standard ResNets on CIFAR-10, CIFAR-100, and ImageNet, with only a modest increase in the training time.

## 2 Background

### 2.1 Backpropagation

Backpropagation [25] is a classic algorithm for computing the gradient of a cost function with respect to the parameters of a neural network. It is used in nearly all neural network algorithms, and is now taken for granted in light of neural network frameworks which implement automatic differentiation [1, 2]. Because achieving the memory savings of our method requires manual implementation of part of the backprop computations, we briefly review the algorithm.

We treat backprop as an instance of reverse mode automatic differentiation [24]. Let $v_1, \ldots, v_K$ denote a topological ordering of the nodes in the network's computation graph $\mathcal{G}$, where $v_K$ denotes the cost function $\mathcal{C}$. Each node is defined as a function $f_i$ of its parents in $\mathcal{G}$. Backprop computes the total derivative $\mathrm{d}\mathcal{C}/\mathrm{d}v_i$ for each node in the computation graph. This total derivative defines the the effect on $\mathcal{C}$ of an infinitesimal change to $v_i$, taking into account the indirect effects through the descendants of $v_k$ in the computation graph. Note that the total derivative is distinct from the partial derivative $\partial f/\partial x_i$ of a function $f$ with respect to one of its arguments $x_i$, which does not take into account the effect of changes to $x_i$ on the other arguments. To avoid using a small typographical difference to represent a significant conceptual difference, we will denote total derivatives using $\overline{v_i} = \mathrm{d}\mathcal{C}/\mathrm{d}v_i$.

Backprop iterates over the nodes in the computation graph in reverse topological order. For each node $v_i$, it computes the total derivative $\overline{v_i}$ using the following rule:

$$\overline{v_i} = \sum_{j \in \mathrm{Child}(i)} \left( \frac{\partial f_j}{\partial v_i} \right)^\top \overline{v_j}, \tag{1}$$

where $\mathrm{Child}(i)$ denotes the children of node $v_i$ in $\mathcal{G}$ and $\partial f_j/\partial v_i$ denotes the Jacobian matrix.

### 2.2 Deep Residual Networks

One of the main difficulties in training very deep networks is the problem of exploding and vanishing gradients, first observed in the context of recurrent neural networks [3]. In particular, because a deep network is a composition of many nonlinear functions, the dependencies across distant layers can be highly complex, making the gradient computations unstable. Highway networks [29] circumvented this problem by introducing skip connections. Similarly, deep residual networks (ResNets) [13] use

a functional form which allows information to pass directly through the network, thereby keeping the computations stable. ResNets currently represent the state-of-the-art in object recognition [13], semantic segmentation [35] and image generation [32]. Outside of vision, residuals have displayed impressive performance in audio generation [31] and neural machine translation [16],

ResNets are built out of modules called residual blocks, which have the following form:

$$y = x + \mathcal{F}(x), \tag{2}$$

where $\mathcal{F}$, a function called the residual function, is typically a shallow neural net. ResNets are robust to exploding and vanishing gradients because each residual block is able to pass signals directly through, allowing the signals to be propagated faithfully across many layers. As displayed in Figure 1, residual functions for image recognition generally consist of stacked batch normalization ("BN") [14], rectified linear activation ("ReLU") [23] and convolution layers (with filters of shape three "C3" and one "C1").

As in He et al. [13], we use two residual block architectures: the basic residual function (Figure 1 right-top) and the bottleneck residual function (Figure 1 right-bottom). The bottleneck residual consists of three convolutions, the first is a point-wise convolution which reduces the dimensionality of the feature dimension, the second is a standard convolution with filter size 3, and the final point-wise convolution projects into the desired output feature depth.

$$a(x) = \text{ReLU}(\text{BN}(x)))$$
$$c_k(x) = \text{Conv}_{k \times k}(a(x))$$

$$\text{Basic}(x) = c_3(c_3(x))$$
$$\text{Bottleneck}(x) = c_1(c_3(c_1(x))) \tag{3}$$

## 2.3 Reversible Architectures

Various reversible neural net architectures have been proposed, though for motivations distinct from our own. Deco and Brauer [8] develop a similar reversible architecture to ensure the preservation of information in unsupervised learning contexts. The proposed architecture is indeed residual and constructed to produce a lower triangular Jacobian matrix with ones along the diagonal. In Deco and Brauer [8], the residual connections are composed of all 'prior' neurons in the layer, while NICE and our own architecture segments a layer into pairs of neurons and additively connect one with a residual function of the other. Maclaurin et al. [21] made use of the reversible nature of stochastic gradient descent to tune hyperparameters via gradient descent. Our proposed method is inspired by nonlinear independent components estimation (NICE) [9, 10], an approach to unsupervised generative modeling. NICE is based on learning a non-linear bijective transformation between the data space and a latent space. The architecture is composed of a series of blocks defined as follows, where $x_1$ and $x_2$ are a partition of the units in each layer:

$$y_1 = x_1$$
$$y_2 = x_2 + \mathcal{F}(x_1) \tag{4}$$

Because the model is invertible and its Jacobian has unit determinant, the log-likelihood and its gradients can be tractably computed. This architecture imposes some constraints on the functions the network can represent; for instance, it can only represent volume-preserving mappings. Follow-up work by Dinh et al. [10] addressed this limitation by introducing a new reversible transformation:

$$y_1 = x_1$$
$$y_2 = x_2 \odot \exp(\mathcal{F}(x_1)) + \mathcal{G}(x_1). \tag{5}$$

Here, $\odot$ represents the Hadamard or element-wise product. This transformation has a non-unit Jacobian determinant due to multiplication by $\exp(\mathcal{F}(x_1))$.

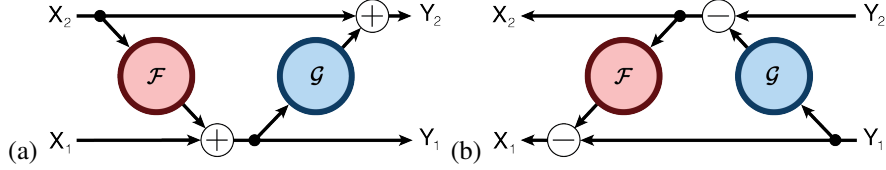

Figure 2: **(a)** the forward, and **(b)** the reverse computations of a residual block, as in Equation 8.

# 3 Methods

We now introduce Reversible Residual Networks (RevNets), a variant of Residual Networks which is reversible in the sense that each layer's activations can be computed from the next layer's activations. We discuss how to reconstruct the activations online during backprop, eliminating the need to store the activations in memory.

## 3.1 Reversible Residual Networks

RevNets are composed of a series of reversible blocks, which we now define. We must partition the units in each layer into two groups, denoted $x_1$ and $x_2$; for the remainder of the paper, we assume this is done by partitioning the channels, since we found this to work the best in our experiments.[2] Each reversible block takes inputs $(x_1, x_2)$ and produces outputs $(y_1, y_2)$ according to the following additive coupling rules – inspired by NICE's [9] transformation in Equation 4 – and residual functions $\mathcal{F}$ and $\mathcal{G}$ analogous to those in standard ResNets:

$$
\begin{aligned}
y_1 &= x_1 + \mathcal{F}(x_2) \\
y_2 &= x_2 + \mathcal{G}(y_1)
\end{aligned}
\tag{6}
$$

Each layer's activations can be reconstructed from the next layer's activations as follows:

$$
\begin{aligned}
x_2 &= y_2 - \mathcal{G}(y_1) \\
x_1 &= y_1 - \mathcal{F}(x_2)
\end{aligned}
\tag{7}
$$

Note that unlike residual blocks, reversible blocks must have a stride of 1 because otherwise the layer discards information, and therefore cannot be reversible. Standard ResNet architectures typically have a handful of layers with a larger stride. If we define a RevNet architecture analogously, the activations must be stored explicitly for all non-reversible layers.

## 3.2 Backpropagation Without Storing Activations

To derive the backprop procedure, it is helpful to rewrite the forward (left) and reverse (right) computations in the following way:

$$
\begin{aligned}
z_1 &= x_1 + \mathcal{F}(x_2) & \qquad z_1 &= y_1 \\
y_2 &= x_2 + \mathcal{G}(z_1) & \qquad x_2 &= y_2 - \mathcal{G}(z_1) \\
y_1 &= z_1 & \qquad x_1 &= z_1 - \mathcal{F}(x_2)
\end{aligned}
\tag{8}
$$

Even though $z_1 = y_1$, the two variables represent distinct nodes of the computation graph, so the total derivatives $\overline{z_1}$ and $\overline{y_1}$ are different. In particular, $\overline{z_1}$ includes the indirect effect through $y_2$, while $\overline{y_1}$ does not. This splitting lets us implement the forward and backward passes for reversible blocks in a modular fashion. In the backwards pass, we are given the activations $(y_1, y_2)$ and their total derivatives $(\overline{y_1}, \overline{y_2})$ and wish to compute the inputs $(x_1, x_2)$, their total derivatives $(\overline{x_1}, \overline{x_2})$, and the total derivatives for any parameters associated with $\mathcal{F}$ and $\mathcal{G}$. (See Section 2.1 for our backprop

notation.) We do this by combining the reconstruction formulas (Eqn. 8) with the backprop rule (Eqn. 1). The resulting algorithm is given as Algorithm 1.[3]

By applying Algorithm 1 repeatedly, one can perform backprop on a sequence of reversible blocks if one is given simply the activations and their derivatives for the top layer in the sequence. In general, a practical architecture would likely also include non-reversible layers, such as subsampling layers; the inputs to these layers would need to be stored explicitly during backprop. However, a typical ResNet architecture involves long sequences of residual blocks and only a handful of subsampling layers; if we mirror the architecture of a ResNet, there would be only a handful of non-reversible layers, and the number would not grow with the depth of the network. In this case, the storage cost of the activations would be small, and independent of the depth of the network.

**Computational overhead.** In general, for a network with $N$ connections, the forward and backward passes of backprop require approximately $N$ and $2N$ add-multiply operations, respectively. For a RevNet, the residual functions each must be recomputed during the backward pass. Therefore, the number of operations required for reversible backprop is approximately $4N$, or roughly 33% more than ordinary backprop. (This is the same as the overhead introduced by checkpointing [22].) In practice, we have found the forward and backward passes to be about equally expensive on GPU architectures; if this is the case, then the computational overhead of RevNets is closer to 50%.

---

**Algorithm 1** Reversible Residual Block Backprop

1: **function** BLOCKREVERSE$((y_1, y_2), (\overline{y}_1, \overline{y}_2))$
2: $\quad z_1 \leftarrow y_1$
3: $\quad x_2 \leftarrow y_2 - \mathcal{G}(z_1)$
4: $\quad x_1 \leftarrow z_1 - \mathcal{F}(x_2)$
5: $\quad \overline{z}_1 \leftarrow \overline{y}_1 + \left(\frac{\partial \mathcal{G}}{\partial z_1}\right)^\top \overline{y}_2$  $\qquad\qquad\qquad\qquad\qquad\qquad$ ▷ ordinary backprop
6: $\quad \overline{x}_2 \leftarrow \overline{y}_2 + \left(\frac{\partial \mathcal{F}}{\partial x_2}\right)^\top \overline{z}_1$  $\qquad\qquad\qquad\qquad\qquad\qquad$ ▷ ordinary backprop
7: $\quad \overline{x}_1 \leftarrow \overline{z}_1$
8: $\quad \overline{w}_\mathcal{F} \leftarrow \left(\frac{\partial \mathcal{F}}{\partial w_\mathcal{F}}\right)^\top \overline{z}_1$  $\qquad\qquad\qquad\qquad\qquad\qquad$ ▷ ordinary backprop
9: $\quad \overline{w}_\mathcal{G} \leftarrow \left(\frac{\partial \mathcal{G}}{\partial w_\mathcal{G}}\right)^\top \overline{y}_2$  $\qquad\qquad\qquad\qquad\qquad\qquad$ ▷ ordinary backprop
10: $\quad$ **return** $(x_1, x_2)$ and $(\overline{x}_1, \overline{x}_2)$ and $(\overline{w}_\mathcal{F}, \overline{w}_\mathcal{G})$
11: **end function**

---

**Modularity.** Note that Algorithm 1 is agnostic to the form of the residual functions $\mathcal{F}$ and $\mathcal{G}$. The steps which use the Jacobians of these functions are implemented in terms of ordinary backprop, which can be achieved by calling automatic differentiation routines (e.g. `tf.gradients` or `Theano.grad`). Therefore, even though implementing our algorithm requires some amount of manual implementation of backprop, one does not need to modify the implementation in order to change the residual functions.

**Numerical error.** While Eqn. 8 reconstructs the activations exactly when done in exact arithmetic, practical `float32` implementations may accumulate numerical error during backprop. We study the effect of numerical error in Section 5.2; while the error is noticeable in our experiments, it does not significantly affect final performance. We note that if numerical error becomes a significant issue, one could use fixed-point arithmetic on the $x$'s and $y$'s (but ordinary floating point to compute $\mathcal{F}$ and $\mathcal{G}$), analogously to [21]. In principle, this would enable exact reconstruction while introducing little overhead, since the computation of the residual functions and their derivatives (which dominate the computational cost) would be unchanged.

## 4 Related Work

A number of steps have been taken towards reducing the storage requirements of extremely deep neural networks. Much of this work has focused on the modification of memory allocation within the training algorithms themselves [1, 2]. Checkpointing [22, 5, 12] is one well-known technique which

Table 1: Computational and spatial complexity comparisons. $L$ denotes the number of layers.

| Technique | Spatial Complexity (Activations) | Computational Complexity |
|---|---|---|
| Naive | $\mathcal{O}(L)$ | $\mathcal{O}(L)$ |
| Checkpointing [22] | $\mathcal{O}(\sqrt{L})$ | $\mathcal{O}(L)$ |
| Recursive Checkpointing [5] | $\mathcal{O}(\log L)$ | $\mathcal{O}(L \log L)$ |
| Reversible Networks (Ours) | $\mathcal{O}(1)$ | $\mathcal{O}(L)$ |

trades off spatial and temporal complexity; during backprop, one stores a subset of the activations (called checkpoints) and recomputes the remaining activations as required. Martens and Sutskever [22] adopted this technique in the context of training recurrent neural networks on a sequence of length $T$ using backpropagation through time [33], storing every $\lceil \sqrt{T} \rceil$ layers and recomputing the intermediate activations between each during the backward pass. Chen et al. [5] later proposed to recursively apply this strategy on the sub-graph between checkpoints. Gruslys et al. [12] extended this approach by applying dynamic programming to determine a storage strategy which minimizes the computational cost for a given memory budget.

To analyze the computational and memory complexity of these alternatives, assume for simplicity a feed-forward network consisting of $L$ identical layers. Again, for simplicity, assume the units are chosen such that the cost of forward propagation or backpropagation through a single layer is 1, and the memory cost of storing a single layer's activations is 1. In this case, ordinary backpropagation has computational cost $2L$ and storage cost $L$ for the activations. The method of Martens and Sutskever [22] requires $2\sqrt{L}$ storage, and it demands an additional forward computation for each layer, leading to a total computational cost of $3L$. The recursive algorithm of Chen et al. [5] reduces the required memory to $\mathcal{O}(\log L)$, while increasing the computational cost to $\mathcal{O}(L \log L)$. In comparison to these, our method incurs $\mathcal{O}(1)$ storage cost — as only a single block must be stored — and computational cost of $3L$. The time and space complexities of these methods are summarized in Table 1.

Another approach to saving memory is to replace backprop itself. The decoupled neural interface [15] updates each weight matrix using a gradient approximation, termed the *synthetic gradient*, computed based on only the node's activations instead of the global network error. This removes any long-range gradient computation dependencies in the computation graph, leading to $\mathcal{O}(1)$ activation storage requirements. However, these savings are achieved only after the synthetic gradient estimators have been trained; that training requires all the activations to be stored.

## 5   Experiments

We experimented with RevNets on three standard image classification benchmarks: CIFAR-10, CIFAR-100, [17] and ImageNet [26]. In order to make our results directly comparable with standard ResNets, we tried to match both the computational depth and the number of parameters as closely as possible. We observed that each reversible block has a computation depth of two original residual blocks. Therefore, we reduced the total number of residual blocks by approximately half, while approximately doubling the number of channels per block, since they are partitioned into two. Table 2 shows the details of the RevNets and their corresponding traditional ResNet. In all of our experiments, we were interested in whether our RevNet architectures (which are far more memory efficient) were able to match the classification accuracy of ResNets of the same size.

### 5.1   Implementation

We implemented the RevNets using the TensorFlow library [1]. We manually make calls to TensorFlow's automatic differentiation method (i.e. `tf.gradients`) to construct the backward-pass computation graph without referencing activations computed in the forward pass. While building the backward graph, we reconstruct the input activations $(\hat{x}_1, \hat{x}_2)$ for each block (Equation 8); Second, we apply `tf.stop_gradient` on the reconstructed inputs to prevent auto-diff from traversing into the reconstructions' computation graph, then call the forward functions again to compute $(\hat{y}_1, \hat{y}_2)$ (Equation 8). Lastly, we use auto-diff to traverse from $(\hat{y}_1, \hat{y}_2)$ to $(\hat{x}_1, \hat{x}_2)$ and the parameters $(w_\mathcal{F}, w_\mathcal{G})$. This

Table 2: Architectural details. 'Bottleneck' indicates whether the residual unit type used was the *Bottleneck* or *Basic* variant (see Equation 3). 'Units' indicates the number of residual units in each group. 'Channels' indicates the number of filters used in each unit in each group. 'Params' indicates the number of parameters, in millions, each network uses.

| Dataset | Version | Bottleneck | Units | Channels | Params (M) |
|---------|---------|------------|-------|----------|------------|
| CIFAR-10 (100) | ResNet-32 | No | 5-5-5 | 16-16-32-64 | 0.46 (0.47) |
| CIFAR-10 (100) | RevNet-38 | No | 3-3-3 | 32-32-64-112 | 0.46 (0.48) |
| CIFAR-10 (100) | ResNet-110 | No | 18-18-18 | 16-16-32-64 | 1.73 (1.73) |
| CIFAR-10 (100) | RevNet-110 | No | 9-9-9 | 32-32-64-128 | 1.73 (1.74) |
| CIFAR-10 (100) | ResNet-164 | Yes | 18-18-18 | 16-16-32-64 | 1.70 (1.73) |
| CIFAR-10 (100) | RevNet-164 | Yes | 9-9-9 | 32-32-64-128 | 1.75 (1.79) |
| ImageNet | ResNet-101 | Yes | 3-4-23-3 | 64-128-256-512 | 44.5 |
| ImageNet | RevNet-104 | Yes | 2-2-11-2 | 128-256-512-832 | 45.2 |

Table 3: Classification error on CIFAR

| Architecture | CIFAR-10 [17] | | CIFAR-100 [17] | |
|--------------|--------|--------|--------|--------|
| | ResNet | RevNet | ResNet | RevNet |
| 32 (38) | **7.14%** | 7.24% | 29.95% | **28.96%** |
| 110 | **5.74%** | 5.76% | 26.44% | **25.40%** |
| 164 | 5.24% | **5.17%** | **23.37%** | 23.69% |

implementation leverages the convenience of the auto-diff functionality to avoid manually deriving gradients; however the computational cost becomes $5N$, compared with $4N$ for Algorithm 1, and $3N$ for ordinary backpropagation (see Section 3.2). The full theoretical efficiency can be realized by reusing the $\mathcal{F}$ and $\mathcal{G}$ graphs' activations that were computed in the reconstruction steps (lines 3 and 4 of Algorithm 1).

Table 4: Top-1 classification error on ImageNet (single crop)

| ResNet-101 | RevNet-104 |
|------------|------------|
| **23.01%** | 23.10% |

## 5.2 RevNet performance

Our ResNet implementation roughly matches the previously reported classification error rates [13]. As shown in Table 3, our RevNets roughly matched the error rates of traditional ResNets (of roughly equal computational depth and number of parameters) on CIFAR-10 & 100 as well as ImageNet (Table 4). In no condition did the RevNet underperform the ResNet by more than 0.5%, and in some cases, RevNets achieved slightly better performance. Furthermore, Figure 3 compares ImageNet training curves of the ResNet and RevNet architectures; reversibility did not lead to any noticeable per-iteration slowdown in training. (As discussed above, each RevNet update is about 1.5-2× more expensive, depending on the implementation.) We found it surprising that the performance matched so closely, because reversibility would appear to be a significant constraint on the architecture, and one might expect large memory savings to come at the expense of classification error.

**Impact of numerical error.** As described in Section 3.2, reconstructing the activations over many layers causes numerical errors to accumulate. In order to measure the magnitude of this effect, we computed the angle between the gradients computed using stored and reconstructed activations over the course of training. Figure 4 shows how this angle evolved over the course of training for a CIFAR-10 RevNet; while the angle increased during training, it remained small in magnitude.

Table 5: Comparison of parameter and activation storage costs for ResNet and RevNet.

| Task | Parameter Cost | Activation Cost |
|---|---|---|
| ResNet-101 | $\sim 178\text{MB}$ | $\sim 5250\text{MB}$ |
| RevNet-104 | $\sim 180\text{MB}$ | $\sim 1440\text{MB}$ |

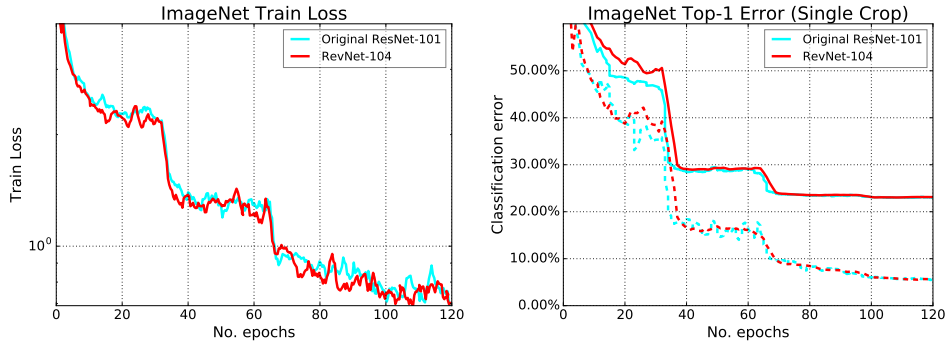

Figure 3: Training curves for ResNet-101 vs. RevNet-104 on ImageNet, with both networks having approximately the same depth and number of free parameters. **Left:** training cross entropy; **Right:** classification error, where dotted lines indicate training, and solid lines validation.

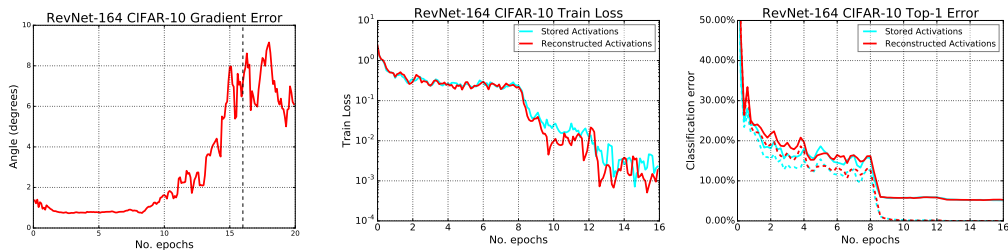

Figure 4: **Left:** angle (degrees) between the gradient computed using stored and reconstructed activations throughout training. While the angle grows during training, it remains small in magnitude. We measured 4 more epochs after regular training length and did not observe any instability. **Middle:** training cross entropy; **Right:** classification error, where dotted lines indicate training, and solid lines validation; No meaningful difference in training efficiency or final performance was observed between stored and reconstructed activations.

Figure 4 also shows training curves for CIFAR-10 networks trained using both methods of computing gradients. Despite the numerical error from reconstructing activations, both methods performed almost indistinguishably in terms of the training efficiency and the final performance.

# 6 Conclusion and Future Work

We introduced RevNets, a neural network architecture where the activations for most layers need not be stored in memory. We found that RevNets provide considerable reduction in the memory footprint at little or no cost to performance. As future work, we are currently working on applying RevNets to the task of semantic segmentation, the performance of which is limited by a critical memory bottleneck — the input image patch needs to be large enough to process high resolution images; meanwhile, the batch size also needs to be large enough to perform effective batch normalization (e.g. [36]). We also intend to develop reversible recurrent neural net architectures; this is a particularly interesting use case, because weight sharing implies that most of the memory cost is due to storing the activations (rather than parameters). Another interesting direction is predicting the activations of previous layers' activation, similar to synthetic gradients. We envision our reversible block as a module which will soon enable training larger and more powerful networks with limited computational resources.

## Footnotes

[2]The possibilities we explored included columns, checkerboard, rows and channels, as done by [10]. We found that performance was consistently superior using the channel-wise partitioning scheme and comparable across the remaining options. We note that channel-wise partitioning has also been explored in the context of multi-GPU training via 'grouped' convolutions [18], and more recently, convolutional neural networks have seen significant success by way of 'separable' convolutions [27, 6].

[3]We assume for notational clarity that the residual functions do not share parameters, but Algorithm 1 can be trivially extended to a network with weight sharing, such as a recurrent neural net.

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

# 7 Appendix

## 7.1 Experiment details

For our CIFAR-10/100 experiments, we fixed the mini-batch size to be 100. The learning rate was initialized to 0.1 and decayed by a factor of 10 at 40K and 60K training steps, training for a total of 80K steps. The weight decay constant was set to $2 \times 10^{-4}$ and the momentum was set to 0.9. We subtracted the mean image, and augmented the dataset with random cropping and random horizontal flipping.

For our ImageNet experiments, we fixed the mini-batch size to be 256, split across 4 Titan X GPUs with data parallelism [28]. We employed synchronous SGD [4] with momentum of 0.9. The model was trained for 600K steps, with factor-of-10 learning rate decays scheduled at 160K, 320K, and 480K steps. Weight decay was set to $1 \times 10^{-4}$. We applied standard input preprocessing and data augmentation used in training Inception networks [30]: pixel intensity rescaled to within [0, 1], random cropping of size $224 \times 224$ around object bounding boxes, random scaling, random horizontal flipping, and color distortion, all of which are available in TensorFlow. For the original ResNet-101, We were unable to fit a mini-batch size of 256 on 4 GPUs, so we instead averaged the gradients from two serial runs with mini-batch size 128 (32 per GPU). For the RevNet, we were able to fit a mini-batch size of 256 on 4 GPUs (i.e. 64 per GPU).

## 7.2 Memory savings

Fully realizing the theoretical gains of RevNets can be a non-trivial task and require precise low-level GPU memory management. We experimented with two different implementations within TensorFlow:

With the first, we were able to reach reasonable spatial gains using "Tensor Handles" provided by TensorFlow, which preserve the activations of graph nodes between calls to `session.run`. Multiple `session.run` calls ensures that TensorFlow frees up activations that will not be referenced later. We segment our computation graph into separate sections and save the bordering activations and gradients into the persistent Tensor Handles. During the forward pass of the backpropagation algorithm, each section of the graph is executed sequentially with the input tensors being reloaded from the previous section and the output tensors being saved for use in the subsequent section. We empirically verified the memory gain by fitting at least twice the number of examples while training ImageNet. Each GPU can now fit a mini-batch size of 128 images, compared the original ResNet, which can only fit a mini-batch size of 32. The graph splitting trick brings only a small computational overhead (around 10%).

The second and most significant spatial gains were made by implementing each residual stack as a `tf.while_loop` with the `back_prop` parameter set to `False`. This setting ensures that activations of each layer in the residual stack (aside from the last) are discarded from memory immediately after their utility expires. We use the `tf.while_loops` for both the forward and backward passes of the layers, ensuring both efficiently discard activations. Using this implementation we were able to train a 600-layer RevNet on the ImageNet image classification challenge on a single GPU; despite being prohibitively slow to train this demonstrates the potential for massive savings in spatial costs of training extremely deep networks.

