[Reviews · NeurIPS 2017]

Reviewer 1



The authors introduce “RevNets”, which avoid storing (some) activations by utilizing computational blocks that are trivial to invert (i.e. y1=x1+f(x2), y2=x2 + g(y1) ). Revnets match the performance of ResNets with the same number of parameters, and in practice RevNets appear to save ~4X in storage at the cost of a ~2X increase in computation. Interestingly, the reversible blocks are also volume preserving, which is not explicitly discussed, but should be, because this is a potential limitation. The approach of reconstructing activations rather than storing them is only applicable to invertible layers, and so while requiring only O(1) storage for invertible layers, succeeds in only a 4X gain in storage requirements (which is nevertheless impressive). One concern I have is that the recent work on decoupled neural interfaces (DNI) is not adequately discussed or compared to (DNI also requires O(1) storage, and estimates error signals [and optionally input values] analogously to how value functions are learned in reinforcement learning). While DNI is not lossless as the authors mention, preliminary indications are that conditional DNI (cDNI, e.g. on labels) is quite effective https://arxiv.org/pdf/1608.05343.pdf, figure 7. DNI has other advantages as well, but indeed is not fully evolved. Nevertheless I think that DNI should be included in table 1, and discussed more thoroughly (If cDNI is highly effective on large scale tasks it would subsume Revnets). Overall, I believe that this paper will be of interest to practitioners in the field, and the idea, while straightforward, is interesting. Minor comments: - Checkpointing is straightforward and general, and may require less overhead, and so should probably be directly compared to at least under the same constraints (store corr. non-invertible layers). More generally, the experiments section should be more explicit wrt the realized memory/compute trade-off. “We empirically verified the memory gain by fitting at least twice the number of examples while training ImageNet” - This confused me---is the gain not 4X? --- Authors: Thank you for your feedback. I've updated my recommendation since 1) DNI is not yet officially published, and 2) the practical memory advantages of RevNets have/will be made clear in the final version of the paper. Good luck!

Reviewer 2



The paper introduces a reversible block for residual networks that has the benefit of not needing to store all of the forward pass activations for the backward pass. This enables training of larger models or minibatches, as the memory footprint during training is smaller. As the memory of GPUs is often a bottleneck when training very large models, the paper is a welcome addition to the line of literature on decreasing this footprint. By training on both CIFAR and ImageNet, the experiments focus on two widely used benchmarks. The experiments seem solid and achieve compelling results. The authors also confirm empirically that numerical instabilities are not a significant problem, which is good. To make the paper even stronger, I would have been interested in seeing even more experiments. It is good that the model is tested both on CIFAR (10 and 100) and ImageNet, but for example SVHN would have been a fairly "cheap" addition to the set of standard benchmarks. The performance on RNNs, that the authors also discuss, would be very interesting to see. In addition, the main concern I have with using the model in practice is the potential drawback of not discarding information (hence the reversibility) - I'm not sure exactly what form that discussion would take, but a slightly deeper discussion on that issue would be helpful. The second paper on NICE might give some ideas for how to do that. A minor issue is also that it is somewhat unclear to the reader how large a typical memory footprint of the activations is compared to the footprint of the weights for a particular minibatch size in the resnets. This obviously depends on many factors, but it would be really helpful (for instance in the appendix) to make some back-of-the-envelope calculations that would give an indication of e.g. how much deeper models or how much larger minibatches one could train, given an optimized code, for typical model sizes. The paper is generally well written, although some minor typos are still present (p1: "simply by widening (increasing the convolutions' filter count) fewer layer" should probably read "...layers", p2: "...derivative defines the the effect..." has one "the" too much, p2: "change to v_i, taking into account the indirect effects through the descendants of v_k" should probably read "... descendants of v_i") which will be easy to fix by carefully rereading the paper a few times. The drawbacks are however fairly minor, and the strength of the paper is that the idea is fairly simple, has been shown with sufficient empirical results to work, and is well presented. In my view, this is an idea that would interest the NIPS audience.